# Learning Domain-Agnostic Representation for Disease Diagnosis

**Churan Wang**[12], **Jing Li**[1], **Xinwei Sun**[7],[*] **Fandong Zhang**[5], **Yizhou Yu**[6], **Yizhou Wang**[234]

1 School of Computer Science, Peking University
2 CFCS, School of CS, Inst. for AI, Peking University
3 Nat'l Key Lab. of GAI & Beijing Institute for GAI (BIGAI)
4 Nat'l Eng. Research Center of Visual Technology
5 AI lab, Deepwise, Beijing, China
6 Department of Computer Science, The University of Hong Kong
7 School of data Science, Fudan University
`{churanwang, lijingg, yizhou.wang}@pku.edu.cn`
`zhangfandong@deepwise.com, sunxinwei@fudan.edu.cn, yizhouy@acm.org`

## Abstract

In clinical environments, image-based diagnosis is desired to achieve robustness on multi-center samples. Toward this goal, a natural way is to capture only clinically disease-related features. However, such disease-related features are often entangled with center-effect, disabling robust transferring to unseen centers/domains. To disentangle disease-related features, we first leverage structural causal modeling to explicitly model disease-related and center-effects that are provable to be disentangled from each other. Guided by this, we propose a novel **D**omain **A**gnostic **R**epresentation **Mo**del (DarMo) based on variational Auto-Encoder. To facilitate disentanglement, we design domain-agnostic and domain-aware encoders to respectively capture disease-related features and varied center effects by incorporating a domain-aware batch normalization layer. Besides, we constrain the disease-related features to well predict the disease label as well as clinical attributes, by leveraging Graph Convolutional Network (GCN) into our decoder. The effectiveness and utility of our method are demonstrated by the superior performance over others on both public datasets and in-house datasets.

## 1 Introduction

A major barrier to the deployment of current deep learning systems to medical imaging diagnosis lies in their non-robustness to distributional shift between internal and external cohorts (Castro et al., 2020; Ma et al., 2022; Lu et al., 2022), which commonly exists among multiple healthcare centers (*e.g.*, hospitals) due to differences in image acquisition protocols. For example, the image appearance can vary a lot among scanner models, parameters setting, and data preprocessing, as shown in Fig. 1 (a,b,c). Such a shift can deteriorate the performance of trained models, as manifested by a nearly 6.7% AUC drop of empirical risk minimization (ERM) method from internal cohorts (source domain, in distribution) to external cohorts (unseen domain, out of distribution), as shown in Fig. 1 (bar graph).

To resolve this problem, existing studies have been proposed to learn task-related features (Castro et al., 2020; Kather et al., 2022; Wang et al., 2021b) from multiple environments of data. Although the learned representation can capture lesion-related information, it is not guaranteed that such features can be disentangled from the *center effect*, *i.e.*, to variations in image distributions due to domain differences in acquisition protocols (Fang et al., 2020; Du et al., 2019; Garg et al., 2021). The mixtures of such variations lead to biases in learned features and final predictions. Therefore, a key question in robustness is: *in which way can the disease-related features be disentangled from center-effect?* Recently, (Sun et al., 2021) showed that the task-related features can be disentangled from others, but requires that the input $X$ and the output $Y$ are generated simultaneously. However, this requirement often does not satisfy disease prediction scenarios, *e.g.*, $Y$ can refer to ground-truth disease labels acquired from pathological examination, which can affect lesion patterns in image $X$.

---

[*]indicates corresponding author

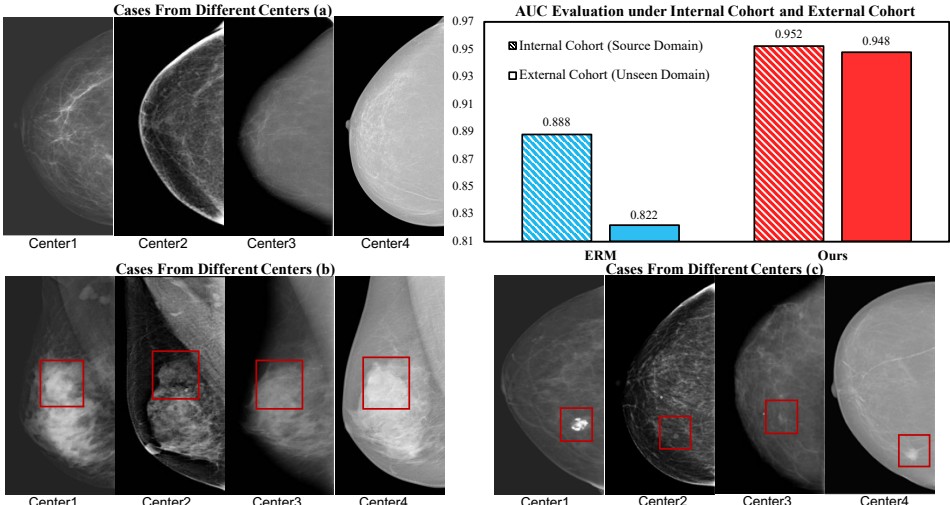

Figure 1: Domain differences between multi centers (Cases-a,b,c) and AUC evaluation of **Ours/** ERM (training by Empirical Risk Minimization) under internal/external cohort. Cases-a,b,c: similar cases in different centers (red rectangles: lesion areas). The bar graph: in the external cohort (unseen domain) ERM performs a large drop on AUC, instead, our proposed method performs stable.

To achieve this disentanglement, we build our model in Fig. 2 (b), via *structural causal modeling* (SCM) that can effectively encode prior knowledge beyond data with hidden variables and causal relations. As shown, we introduce $v_{\text{ma}}$ and $v_{\text{mi}}$ to respectively denote *macroscopic* and *microscopic* parts of disease-related features that often employed in clinical diagnosis. Specifically, the macroscopic features encode morphology-related attributes (Surendiran & Vadivel, 2012) of lesion areas, as summarized in American College of Radiology (ACR) (Sickles et al., 2013); while the microscopic features are hard to observe but reflect subtle patterns of lesions. Taking the mammogram in Fig. 2 (a) as an illustration, the macroscopic features refer to the margins, shapes, and speculations of the masses; while the microscopic features refer to the textures, and the curvatures of contours (Ding et al., 2020a). As these disease-related patterns vary between malignancy and benign, they are determined by the disease status $Y$ and we have $y \rightarrow (v_{\text{ma}}, v_{\text{mi}})$ in Fig. 2 (b) correspondingly. Besides, the $v_{\text{ma}}$ differs from $v_{\text{mi}}$, as it is related to clinical attributes $A$ that are easy to observe from the image. In addition to disease-related features, we also introduce $v_d$ to account for domain gaps from the center effect in the image. Note that given the image $X$ (*i.e.*, condition on $X$), the $v_d$ is correlated to $(v_{\text{ma}}, v_{\text{mi}})$, making them entangled with each other. This entanglement can cause bias and thus unstable prediction behaviors when transferred to unseen centers/domains.

Equipped with this causal modeling, we can observe that the distributional shift of data is mainly accounted for by the variation of $v_d$ across domains. Moreover, we can theoretically prove that when this variation is diverse enough, the disease-related features can be disentangled from the center effect. To the best of our knowledge, we are the first to prove that this disentanglement is possible, in the literature on imaging diagnosis. Inspired by this result, we propose a disentangling learning framework, dubbed as **D**omain **A**gnostic **R**epresentation **Mo**del (DarMo), to disentangle disease-related features for prediction. Specifically, we adopt a variational auto-encoder framework and decompose the encoder into domain-agnostic and domain-aware branches, which respectively encode disease-related information $(v_{\text{ma}}, v_{\text{mi}})$ and domain effect $v_d$. To account for the variation of $v_d$ across domains, we propose to incorporate a domain-aware batch normalization (BN) layer into the domain-aware encoder, to well capture the effect in each domain. To capture disease-related information, we use disease labels to supervise $(v_{\text{ma}}, v_{\text{mi}})$ and additionally constrain $v_{\text{ma}}$ to reconstruct clinical attributes with Graph Convolutional Network (GCN) to model relations among attributes.

To verify the utility and effectiveness of our method, we perform our method on mammogram benign/malignant classification. Here the clinical attributes are those related to the masses, which are summarized in ACR (Sickles et al., 2013) and are easy to obtain. We consider four datasets (one public and three in-house) that are collected from different sources. The results on unseen domains show that our method can outperform others by 6.2%. Besides, our learned disease-related features can successfully encode the information on the lesion areas.

In summary, our contributions are mainly three-fold: **a)** We leverage SCM to encode medical prior knowledge, equipped with which we theoretically show that the disease-related features can be disentangled from the domain effect; **b)** We propose a novel DarMO framework with domain-agnostic and domain-aware encoders, which facilitates the disentanglement of disease-related features from center effect to achieve robust prediction; **c)** Our model can achieve state-of-the-art performance in terms of robustness to distributional shifts across domains in breast cancer diagnosis.

## 2 RELATED WORK

**Domain-Agnostic Representation Learning for Disease Diagnosis.** The multi-center study is important for clinical diagnosis Liu et al. (2020); Kather et al. (2022); Castro et al. (2020); Pollard et al. (2018). It considers multiple domains (centers) and aims to improve the diagnosis performance in unseen domains. However, under unseen domains, previous methods will lead to a dramatic performance decrease when testing on data from a different domain with a different bias (Ilse et al., 2020; Sathitratanacheewin et al., 2020; Zhang et al., 2022a). Thus such previous models are not robust enough to the actual task (Azulay & Weiss, 2020; Cheng et al., 2022). An intuitive idea to solve domain gaps among multi-centers is learning domain-agnostic representation. Progress has been made can be roughly divided into three classes: (i) Learning the domain-specific constraints, *e.g.,* (Chattopadhyay et al., 2020) aim to learn domain-specific masks but fails in medical images for not suitable to distinguish different domains based on masks. (ii) Disentangle-based, *e.g.,* (Ilse et al., 2020) model three independent latent subspaces for the domain, the class, and the residual variations respectively. They do not make use of the medical attribute knowledge which is important in our mammogram classification. (iii) Design invariant constraints, *e.g.,* (Arjovsky et al., 2019; Zhang et al., 2022b) aim to learn invariant representation across environments by minimizing the Invariant Risk Minimization term. (Ganin et al., 2016) and (Li et al., 2018) use an adversarial way with the former performing domain-adversarial training to ensure a closer match between the source and the target distributions. Lack of disentanglement and the guidance of medical prior knowledge limits their performance on unseen domains.

## 3 METHODOLOGY

**Problem Setup & Notations.** Denote $X \in \mathcal{X}, Y \in \mathcal{Y}, A \in \mathcal{A}$ respectively as the image, benign/malignant label, and clinical attributes. We here formulate different centers as different domains and collect datasets $\{x_i^d, y_i^d, A_i^d\}_{i=1}^{n_d}$ from multiple domains (healthcare centers) $d \in \mathcal{D}$. Our goal is to identify causal features from training domains (centers) $\mathcal{D}_{\text{Tr}}$, so that the induced disease predictor can generalize well to new unseen domains (external cohort) $\mathcal{D}_{\text{Te}}$. Let $m := |\mathcal{D}_{\text{Tr}}|$ be the number of training domains (centers) and $n := \sum_{i=1} n_D$ be the total training samples.

In the following, we first introduce our causal model that incorporates medical priors regarding heterogeneous data from multiple domains in Sec. 3.1. With this modeling, we show that the domain-agnostic causal features can be disentangled from domain-aware features if we can fit distributions of each domain well. Guided by this result, we in Sec. 3.2 propose a variational auto-encoder (VAE)-based method as a generative model to fit these distributions, so as to learn causal features for disease prediction.

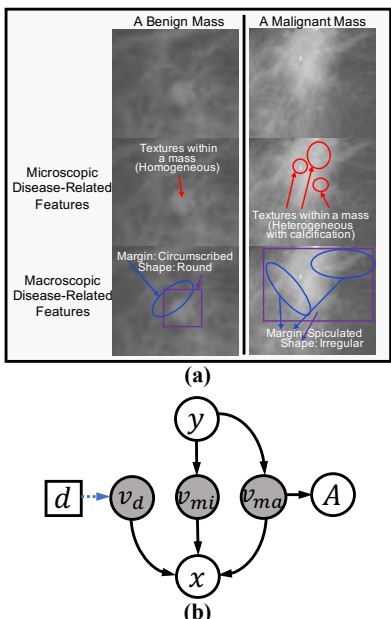

**(a)**

**(b)**

Figure 2: Causal Graph of our model. For $v_{ma}, v_{mi}, v_d$ that respectively denote macroscopic, microscopic, and center-dependent features, the $(v_{ma}, v_{mi})$ are associated with the disease status $y$ and $v_d$ is affected by domain variable $d$.

### 3.1 DOMAIN-AGNOSTIC CAUSAL MODEL FOR MEDICAL IMAGE

We first formally define the notion of *domain-aware features* and *domain-agnostic casual features*, in the framework of our proposed structural causal model in Fig. 2 (b). In this model, we roughly

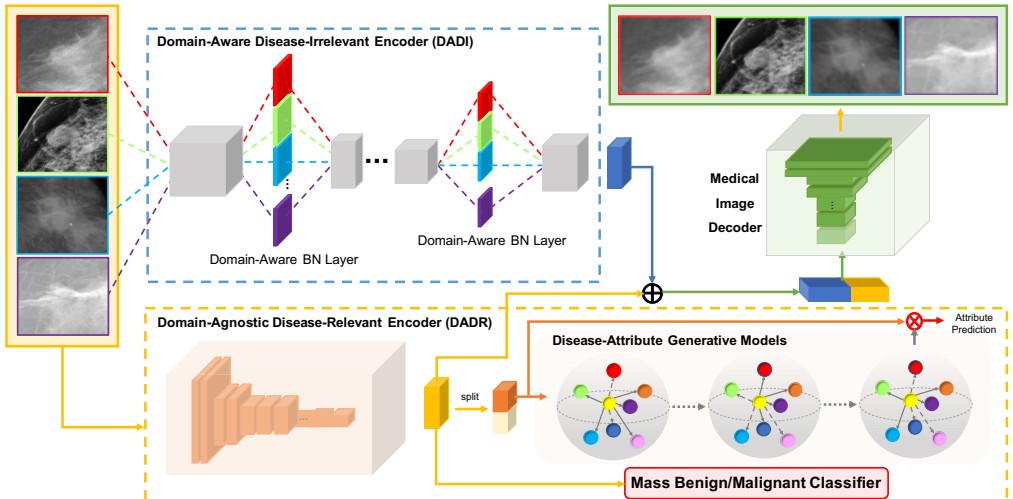

Figure 3: Overview of our VAE-based method, which is composed of two-branch encoder: *Domain-Agnostic Disease-Relevant Encoder* (DADR) to extract macroscopic features $v_{ma}$, microscopic features $v_{mi}$, and *Domain-Aware Disease-Irrelevant Encoder* (DADI) to extract domain-specific effects $v_d$. In DADI, images from different centers are fed into corresponding domain-aware layers respectively, to model the variation of domain effects. In DADR, we implement graph convolution (Disease-Attribute Generative Model) to capture relations among clinical attributes.

decompose latent factors $v$ of the input image $x$ into domain-agnostic causal features $(v_{ma}, v_{mi})$ that are determined by the disease status $y$, and other domain-aware features $v_d$ affected by the domain variable $d$. For domain-agnostic casual features, we further denote $v_{ma}$ as macroscopic features that generate clinical attributes $v_{ma}$ (such as shapes, margins (Sickles et al., 2013; Wang et al., 2021b; Zhao et al., 2022)) that are normally utilized by clinicians for disease prediction, and $v_{mi}$ as microscopic features (such as textures, curvatures of contours (Ding et al., 2020a)) that may be difficult to observe but can encode the high-frequency patterns of lesions. For $v_d$, it can encode biases introduced during the imaging acquisition process from different centers/medical devices.

If we directly train a predictor $p(y|x)$ using a neural network, the extracted representation from $x$ can entangle the causal features $(v_{ma}, v_{mi})$ and center effects $v_d$ because conditioning on $x$ can induce the spurious path from $v_d$ to $(v_{ma}, v_{mi})$, making $v_d$ and $(v_{ma}, v_{mi})$ correlated with each other. Such an entanglement makes it hard to generalize well on new centers' data. Specifically, if we denote $S$ as the learned representation from training domains' data, then $S$'s distribution of the diseased group can be affected by $v_d$, which is domain-aware. Therefore, this distribution can change a lot on another domain's data, which may cause difficulty in discriminating the diseased group from the normal one in terms of $S$'s distribution.

To remove this domain dependency, it is desired to disentangle causal features from domain-aware features. Indeed, this disentanglement can be achieved via acquisition from multiple domains with diverse distributions. Specifically, the difference between $(v_{ma}, v_{mi})$ and $v_d$ mainly lies in whether this feature is domain-invariant. The diversity among domains can thus provide a clue to identify invariant information, *i.e.*, $(v_{ma}, v_{mi})$ in our scenario, as shown in the following theorem:

**Theorem 3.1** (Informal). *Suppose that multiple domains are diverse enough. Then as long as we can fit each domain's distribution well, then for each image $x \leftarrow f_x(v_{mi}^\star, v_{ma}^\star, v_c^\star)$, the learned factors $(\tilde{v}_{mi}, \tilde{v}_{ma}, \tilde{v}_d)$ has $\tilde{v}_{mi} = h_{mi}(v_{mi}^\star)$, $\tilde{v}_d = h_d(v_d^\star)$, $\tilde{v}_{ma} = h_{ma}(v_{ma}^\star)$ for some $h_{mi}, h_{ma}, h_d$.*

**Remark 3.1.** *The diversity condition means the extent of dependency $v_d$ on $d$ and $(v_{ma}, v_{mi})$ on $y$ are large enough, which can be shown to hold generically in the appendix.*

This theorem informs that as long as we can fit data well, we can identify each factor, particularly domain-agnostic causal features $(v_{ma}, v_{mi})$ up to the transformation that does not depend on $v_d$. In this regard, the learned domain-agnostic causal features are disentangled from domain effects. Guided by this analysis, we propose a variational auto-encoder (VAE)-based method, as a generative model to fit data from each center.

### 3.2 GENERATIVE MODELING FOR DOMAIN-AGNOSTIC CAUSAL FEATURES

In this section, we first obtain the objective function via reformulating the **E**vidence **L**ower **BO**und (ELBO) in VAE, which are composed of **Domain-Aware Prior Models**, **Domain-Aware/Agnostic Inference Models** to learn latent factors, and **Disease-Attribute Generative Models** to reconstruct $X, A$ and to predict $y$.

**Objective Function of VAE.** To learn $p^d(x, y, A)$ for each domain $d$, we decompose the whole log-likelihood into $\log p_\theta^d(x, y, A) = \log p_\theta^d(x) + \log p_\theta^d(y, A|x)$, in which the term $\log p_\theta^d(x)$ can be replaced by the ELBO of VAE (Kingma & Welling, 2013), *i.e.*, $\mathbb{E}_{q_{\psi^d}(v|x)}\left(\log \frac{p_\theta^d(x,v)}{q_{\psi^d}(v|x)}\right)$ that is $\leq \log p_\theta^d(x)$. Here, the "=" can only be achieved once the variational distribution $q_{\psi^d}(v|x)$ equals to the ground-truth posterior $p_\theta^d(v|x)$. In this case, we have $p_\theta^d(y, A|x) = \int p_\theta(A|v_{ma})p_\theta(y|v_{ma}, v_{mi})q_{\psi^d}(v_{ma}, v_{mi}, v_d|x)dv_{mi}dv_{ma}dv_d$, since $A \perp v_{mi}, v_d|v_{ma}$ and $y \perp v_d|(v_{ma}, v_{mi})$ according to Fig. 2(b). The empirical loss for data in domain $d$ is:

$$\ell^d(q^d, p_\theta^d) = -\frac{1}{n_d}\sum_{i=1}^{n_d}\left(\log p_\theta^d(y_i, A_i|x_i) + \mathbb{E}_{q_{\psi^d}(v|x_i)}\left(\log \frac{p_\theta^d(x_i, v)}{q_{\psi^d}(v|x_i)}\right)\right),$$

$$= -\frac{1}{n_d}\sum_{i=1}^{n_d}\left(\log\left(\mathbb{E}_{q_{\psi^d}(v|x_i)}\left(p_\theta(A_i|v_{ma})p_\theta(y_i|v_{ma}, v_{mi})\right)\right)\right.$$

$$\left. + \mathbb{E}_{q_{\psi^d}(v|x_i)}\left(\log p_\theta(x_i|v)\right) - \mathrm{KL}(q_{\psi^d}(v|x_i), p_\theta^d(v))\right). \tag{1}$$

with $q_{\psi^d}(v|x)$ learned to approximate $p_\theta^d(v|x)$. To optimize the loss, we need to respectively parameterize the prior models $p_\theta^d(v_{ma}, v_{mi}, v_d) := p_\theta(v_d|d)p_\theta(v_{ma}, v_{mi})$, inference models $q_{\psi^d}(v|x)$ (*i.e.*, encoder) and generative models $p_\theta(x|v_{ma}, v_{mi}, v_d)$, $p_\theta(A|v_{ma})$, $p_\theta(y|v_{ma}, v_{mi})$ (*i.e.*, decoder). In the following, we will introduce our implementation for these models. As illustrated in Fig. 3, we propose a two-branch encoder: *Domain-Agnostic Encoder* to extract $(v_{ma}, v_{mi})$ and *Domain-Aware Encoder* to extract $v_d$. For the latter, we incorporate a *domain-aware BN* to capture the variation of multiple domains. With learned causal features, we implement *graph convolution network* to capture relations among clinical attributes.

**Domain-Aware Prior Models.** Following the causal graph in Fig. 2, we factorize $p_\theta^d(v_{ma}, v_{mi}, v_d)$ into $p_\theta^d(v_{ma}, v_{mi}, v_d) = p(v_{ma}, v_{mi})p_{\theta_{pri}}(v_d|d)$, where the $p(v_{ma}, v_{mi})$ can be modeled as isotropic Gaussian while $p_{\theta_d}(v_d|d)$ is domain-aware, and is parameterized as a Multilayer Perceptron (MLP) with one-hot encoded vector $d \in \mathbb{R}^m$ as input.

**Domain-Aware/Agnostic Inference Models.** To disentangle causal features $(v_{ma}, v_{mi})$ from domain effects, we adopt a mean-field approximation to factorize $q_{\psi^d}(v_d, v_{ma}, v_{mi}|x)$ as $q_{\psi_1}(v_{ma}, v_{mi}|x) * q_{\psi_2^d}(v_d|x)$, with $q_\psi(v_{ma}, v_{mi}|x)$ and $q(v_d|x, d)$ respectively implemented via a *domain-agnostic disease-relevant encoder (DADR)* and a *domain-aware disease-irrelevant encoder (DADI)*. This parameterization is inspired by the domain-invariant/-variant properties of $(v_{ma}, v_{mi})$ and $v_d$. By attributing the domain-aware effects to feature $v_d$ while sharing parameters of the domain-agnostic encoder $\psi_1$ for all centers, the domain-aware effects can be removed in learned macroscopic and microscopic information, leading to robust generalization ability across domains.

With shared parameters of the domain-agnostic encoder, we have

$$p_\theta^d(y|A, x) = \int p_\theta(A|v_{ma})p_\theta(y|v_{ma}, v_{mi})q_\psi(v_{ma}, v_{mi}|x)dv_{mi}dv_{ma}. \tag{2}$$

which hence does not depend on the domain index $d$. To reflect the variety of different domain effects, the domain-aware encoder contains a Domain-Aware Layer (DAL), which is composed of $m$ batch-normalization (BN) layers with $(\gamma_d, \beta_d)$ for each center: $f_d = \mathrm{BN}_{\gamma_i, \beta_i}(\hat{f}) = \gamma_d\hat{f} + \beta_d$, with $\hat{f} = \frac{f - \mu_B}{\sqrt{\delta_B^2 + \epsilon}}$ denoting the normalized features by the mini-batch mean $\mu_B$ and variance $\delta_B$.

**Disease-Attribute Generative Models.** To learn $v_{ma}, v_{mi}, v_d$, we constrain them to well recover $x$ and predict $A, y$, respectively via $p_{\theta_x}(x|v)$, $p_{\theta_y}(y|v_{ma}, v_{mi})$ and $p_{\theta_A}(A|v_{ma})$. Specifically, to capture macroscopic patterns in $v_{ma}$, we constrain it to estimate the clinical attributes $A$ that include macroscopic information such as shape, margins, lobulation, *etc.* As correlations among clinical attributes can be helpful for disease diagnosis, we propose to reconstruct $A$ via Graph Convolutional

Table 1: AUC evaluation of public/in-house datasets on **external cohorts** (unseen domains), *i.e.*, training and testing data are from different domains. ◯: domains for testing, ●: domains for training).

| | | | | |
|---|---|---|---|---|
| InH1 | ◯ | ● | ● | ● |
| InH2 | ● | ◯ | ● | ● |
| InH3 | ● | ● | ◯ | ● |
| DDSM | ● | ● | ● | ◯ |
| ERM (He et al., 2016) | 0.822 | 0.758 | 0.735 | 0.779 |
| (Chen et al., 2019) | 0.877 | 0.827 | 0.804 | 0.830 |
| Guided-VAE (Ding et al., 2020b) | 0.872 | 0.811 | 0.779 | 0.811 |
| IAIA-BL (Barnett et al., 2021) | 0.861 | 0.803 | 0.767 | 0.782 |
| ICADx (Kim et al., 2018) | 0.882 | 0.802 | 0.777 | 0.826 |
| (Li et al., 2019) | 0.848 | 0.794 | 0.769 | 0.815 |
| DANN (Ganin et al., 2016) | 0.857 | 0.811 | 0.781 | 0.813 |
| MMD-AAE (Li et al., 2018) | 0.860 | 0.783 | 0.770 | 0.786 |
| DIVA (Ilse et al., 2020) | 0.865 | 0.809 | 0.784 | 0.813 |
| IRM (Arjovsky et al., 2019) | 0.889 | 0.830 | 0.795 | 0.829 |
| (Chattopadhyay et al., 2020) | 0.851 | 0.796 | 0.772 | 0.797 |
| DDG (Zhang et al., 2022b) | 0.867 | 0.811 | 0.778 | 0.802 |
| EFDM (Zhang et al., 2022c) | 0.864 | 0.812 | 0.765 | 0.796 |
| **Ours** | **0.948** | **0.874** | **0.858** | **0.892** |
| Ours (2/3 DAL) | 0.946 | 0.874 | 0.853 | 0.889 |
| Ours (1/2 DAL) | 0.942 | 0.871 | 0.847 | 0.883 |
| Ours (1/3 DAL) | 0.930 | 0.863 | 0.842 | 0.871 |
| Ours (one layer DAL) | 0.926 | 0.857 | 0.835 | 0.864 |
| Ours (DAL -> ME) | 0.946 | 0.873 | 0.855 | 0.891 |
| Ours (DAL -> GL) | 0.947 | 0.872 | 0.854 | 0.887 |

Network (GCN) Kipf & Welling (2016) that is flexible to capture the topological structure in the label space. More details are left in the appendix.

**Training and Testing.** With above parameterizations, we optimize prior parameters $\theta_{pri} := \{\theta_d\}$, inference parameters $\psi := \{\psi^d\}$ with $\psi^d := (\psi_1, \psi_2^d)$ such that $\psi_2^d$ includes $(\gamma_d, \beta_d)$ and other layers' parameters that do not depend on $d$, and generative parameters $\theta_{gen} := (\theta_x, \theta_y, \theta_A)$ via $\mathcal{L}(\theta_{pri}, \psi, \theta_{gen}) := \sum_d \ell^d(\theta_d, \psi^d, \theta_{gen})$ with $\ell^d$ defined in Eq. 1. During testing stage for a new image $x$, we first extract causal features $(v_{ma}, v_{mi})$ from $x$, followed by prediction via $p_{\theta_y}(y|v_{ma}, v_{mi})$.

## 4 EXPERIMENTS

**Datasets and Implementation.** To evaluate the effectiveness of our model, we apply our model on mammogram mass benign/malignant classification, which drives increasing attention recently (Wang et al., 2021a; Zhao et al., 2018; Wang et al., 2021b; Lei et al., 2020; Wang et al., 2020; 2022) due to its clinical use. **Public dataset (DDSM** (Bowyer et al., 1996)) and **three in-house datasets (InH1, InH2, and InH3)** what we use are from different centers (center4, 1, 2, 3 respectively). Different medical devices, different regions/countries, and different image formats cause domain gaps. For each dataset, we randomly split it into training, validation, and testing sets with an 8:1:1 patient-wise ratio. The inputs of the network are resized into 224 × 224 with random horizontal flips and fed into networks. To verify the effectiveness of multi-center benign/malignant diagnosis, we show our performances on the external cohort (unseen domains) in Tab. 1 (training data and testing data are from different domains). To remove the randomness, we run for 10 times and report their average values. To further validate our effectiveness, we also give internal cohort (source domain,*i.e.*, the same domain as training domain) results of each dataset which can be seen as the upper bounds of each dataset. For a fair comparison, the number of above-all training sets all keep the same. Area Under the Curve (AUC) is used as the evaluation metric image-wise. More details of datasets and implementation are shown Appendix.

### 4.1 RESULTS

**Compared Baselines.** We compare our model with the following methods: **a)** ERM (He et al., 2016) directly trains the classifier via ResNet34 by Empirical Risk Minimization; **b)** Chen *et al.* (Chen et al., 2019) achieves multi-label classification with GCN for attributes prediction; **c)** Guided-VAE

Table 2: AUC evaluation of public/in-house datasets on **internal cohorts** (source domains, *i.e.*, in-distribution: training and testing data are from the same domains. −: do not use, ⊙: the same domain for training and testing).

| | | | | |
|---|---|---|---|---|
| InH1 | ⊙ | - | - | - |
| InH2 | - | ⊙ | - | - |
| InH3 | - | - | ⊙ | - |
| DDSM | - | - | - | ⊙ |
| ERM (He et al., 2016) | 0.888 | 0.847 | 0.776 | 0.847 |
| (Chen et al., 2019) | 0.924 | 0.878 | 0.827 | 0.871 |
| Guided-VAE (Ding et al., 2020b) | 0.921 | 0.867 | 0.809 | 0.869 |
| ICADx (Kim et al., 2018) | 0.911 | 0.871 | 0.816 | 0.879 |
| (Li et al., 2019) | 0.908 | 0.859 | 0.828 | 0.875 |
| IAIA-BL (Barnett et al., 2021) | 0.910 | 0.865 | 0.821 | 0.856 |
| **Ours-single** | **0.952** | **0.898** | **0.864** | **0.919** |

(Ding et al., 2020b) also implements disentangle network but lacks the medical prior knowledge of attributes during learning; **d)** Li *et al.* (Li et al., 2019) improve performance by generating more benign/malignant images via adversarial training; **e)** ICADx (Kim et al., 2018) also proposes the adversarial learning method but additionally introduces shape/margins information for reconstruction; **f)** DANN (Ganin et al., 2016) uses an adversarial way to ensure a closer match between the source and the target distributions; **g)** MMD-AAE (Li et al., 2018) extends adversarial autoencoders by imposing the Maximum Mean Discrepancy (MMD) measure; **h)** DIVA (Ilse et al., 2020) proposes a generative model with three independent latent subspaces; **i)** IRM (Arjovsky et al., 2019) designs Invariant Risk Minimization term to learn invariant representation across environments; **j)** Prithvijit *et al.*(Chattopadhyay et al., 2020) add domain-specific masks learning for better domain generalization. **k)**IAIA-BL (Barnett et al., 2021) diagnoses based on pixel-level prediction. **l)** DDG (Zhang et al., 2022b) formulates a constrained optimization to solve domain generalization. **m)** EFDM (Zhang et al., 2022c) matches empirical distribution functions.

**Results & Analysis on external cohorts (unseen domains).** To verify the effectiveness of our method under unseen domains (out-of-distribution), we train our model on the combination of three datasets from three different centers and test on the external cohort (another unseen dataset from other centers). As shown in Tab. 1 (Lines 1-18), our methods can achieve state-of-the-art results under unseen domains in all settings.

Specifically, the first six lines are the methods based on different representation learning and we extend them to our domain generalization task. The next seven lines are the methods aiming at domain generalization. (Li et al., 2019) generates more data under the current domain, the larger number of data improves the performance compared with ERM (He et al., 2016) but the augmentation for the current domain greatly limits its ability of domain generalization. (Chattopadhyay et al., 2020) learns domain-specific masks (Clipart, Sketch, Painting), however, the gap that exists in medical images can not balance through mask learning. DANN (Ganin et al., 2016), DDG (Zhang et al., 2022b), EFDM (Zhang et al., 2022c) and MMD-AAE (Li et al., 2018) design distance constraints between the source and the target distributions. However, simple distance constraints are not the key to cancer diagnosis and are not robust enough. The advantage of Guided-VAE (Ding et al., 2020b) and DIVA (Ilse et al., 2020) over mentioned methods above may be due to the disentanglement learning in the former methods. IRM (Arjovsky et al., 2019) learns invariant representation across environments by Invariant Risk Minimization. However, lacking the guidance of the disentanglement learning limits their performance. Guided-VAE (Ding et al., 2020b) introduces the attribute prediction which improves their performance more than DIVA (Ilse et al., 2020). The improvements in ICADx (Kim et al., 2018), Guided-VAE (Ding et al., 2020b) prove the importance of the guidance of attribute learning. Although ICADx (Kim et al., 2018) uses the attributes during learning, it fails to model correlations between attributes and diagnosis, which limits their performance. With further exploration of attributes via GCN, our method can outperform ICADx (Kim et al., 2018), Guided-VAE (Ding et al., 2020b). Compared to (Chen et al., 2019) and IAIA-BL (Barnett et al., 2021) that also implement attribute learning, we additionally employ disentanglement learning with variance regularizer which can help to identify invariant disease-related features during prediction.

**Comparisons on internal cohorts (source domains).** We further compute the in-distribution AUC performance of every single dataset under internal cohorts (Tab. 2). Our method shows stable performance while other methods drop a lot under external cohorts compared with Tab. 1.

Table 3: Overall Prediction Accuracy (ACC) of Multi Attributes (Mass shapes, Mass margins) on **external cohorts** (unseen domains, *i.e.*, out-of-distribution: training and testing data are from different domains). Testing names are noted in the table.

| Methodology | InH1 | InH2 | InH3 | DDSM |
|---|---|---|---|---|
| ERM-multitask | 0.618 | 0.560 | 0.596 | 0.662 |
| Chen *et al.* (Chen et al., 2019) | 0.827 | 0.795 | 0.748 | 0.842 |
| ICADX (Kim et al., 2018) | 0.743 | 0.647 | 0.612 | 0.739 |
| Proposed Method | **0.914** | **0.877** | **0.858** | **0.934** |

Specifically, we implement the methods which aim at representation learning on internal cohorts, *i.e.*, training and testing on the data from the same domain. Such in-distribution results can serve as the upper bounds of our generalization method. To adapt our proposed mechanism to the in-distribution situation, we change our network with two branches without domain-aware BN layers accordingly for extracting features into $a, s, z$ since training data is only from one center(**Ours-single**), *i.e.,* one domain without domain influence. As shown in Tab. 2, based on the disentanglement mechanism and the guidance of attribute learning, **Ours-single** still gets the state-of-art performance. We argue that the disentangling mechanism with the guidance of attributes helps effective learning of disease-related features under a single domain. Results in Tab. 2 can be seen as the upper bound results of each setting in Tab. 1. Our results in Tab. 1 are slightly lower than results in Tab. 2 by 0.4% to 2.7%. We argue that based on our mechanism for domain generalization, our method can get evenly matched performance compared under external cohorts (out-of-distribution) with internal cohorts (in-distribution). For example, as shown when testing on DDSM (Bowyer et al., 1996), performances of our model training on InH1+InH2+InH3 and training on DDSM itself are comparable.

## 4.2 ABLATION STUDY

**Ablation study on each component.** To verify the effectiveness of each component in our DarMo, we evaluate some variant models on external cohorts as shown in our appendix-Tab.4.

To abate the impact of the combination of training domains, we also train our model under different training combinations and show results in Appendix. Results indicate that influences between different domains are not obvious and three domains are sufficient to achieve comparable results.

**Ablation study on the ratio of using domain-aware layers.** To verify the effectiveness of the ratio of using DAL, we replaced the original BN layer with DAL in different ratios. The results are shown in Tab. 1 **Line18-21**, specifically, $1/3$ means only $1/3$ BN layers in the network are replaced, others, and so forth. As shown, under the lower ratio, the performance drops due to the poorer domain interpretability. The higher ratio can get better performance.

**Ablation study on Domain-Aware Mechanism** To deeply investigate the proposed domain-aware BN layer, we analyze various implementation forms of multiple domains as follows: **a)** *Multiple Encoders(ME)*. Since the irrelevant encoder contains the information of domain environments, an intuitive idea is to use multiple irrelevant encoders so as to each domain has one irrelevant encoder directly. **b)** *Grouped Layer(GL)*. To reduce the parameter quantity of ME, we consider several groups of blocks with each group containing two blocks in the same structures. Each group only responds to one block each time, and different domains are different block combinations. The number of groups is set to $n$ that satisfies $2^n = m$ ($m$ denotes the number of domains, if $m$ is not the exponential power of 2, find $\hat{m}$ that is larger than $m$ and is the least number that satisfies $2^n = \hat{m}$). Thus each domain is a permutation combination based on each group choosing one block. **c)** *Domain-Aware BN Layer(DAL)*. To further reduce the parameter quantity and achieve domain generalization, we propose the domain-aware BN layer for each domain. The scaling and shifting parameters in each layer are learned adaptively.

We conduct experiments under the mechanisms above and the results are shown in Tab. 1 **Line18-23**. Three different kinds of mechanisms have comparable performance. Since BN can usually be used as an effective measure for domain adaptation (Ioffe & Szegedy, 2015), **DAL** can be slightly better than the others with lighter computation, especially compared to **ME**.

## 4.3 PREDICTION ACCURACY OF ATTRIBUTES

We argue that attributes can be the guidance of benign/malignant classification. In the current domain generalization task, under external cohorts (unseen domain), we also calculate the prediction accuracy

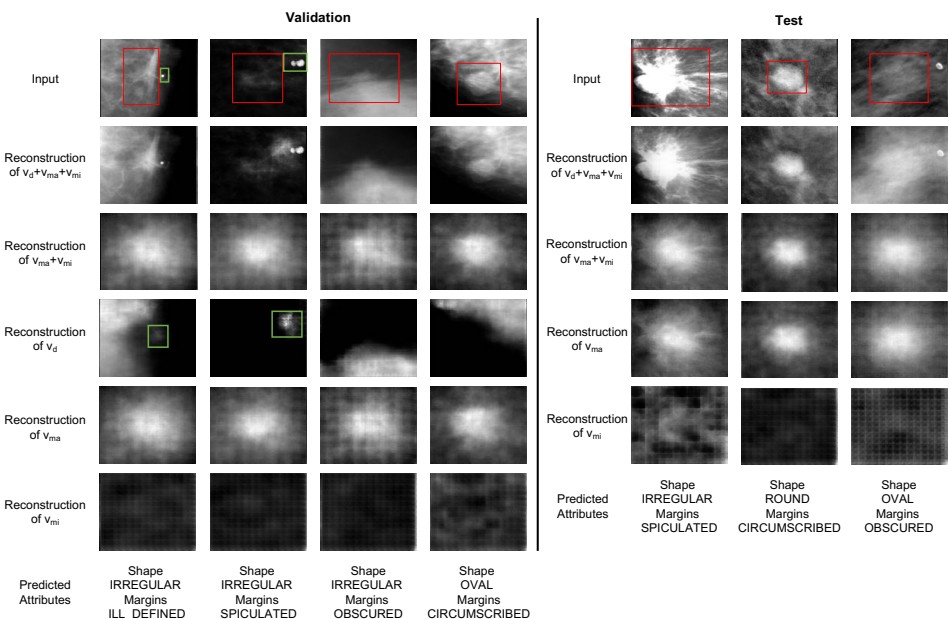

Figure 4: Visualization on valid(internal) and test(external) cohorts. Red rectangles: lesion regions; green rectangles: white dots caused by machine shooting. Each row: the reconstruction of different latent variables. Validation: 1st and 4th columns are from the center2, the 2nd column is from the center1, and the 3rd column is from the center3. Test: All columns are from center 4. Note that there are no reconstruction results of $v_d$ at the test stage because the test domains have no corresponding domain-aware encoders.

of attributes in ours and other attribute-based representative methods in Tab. 3. Our method gets the best prediction accuracy on the attributes over other methods under out-of-distribution.

## 4.4 VISUALIZATION

We visualize reconstruction results of all latent factors and the predicted attributes of the current image in Fig. 4 to validate that our model can successfully disentangle latent factors $v_{ma}$, $v_{mi}$, and $v_d$. Since the *DADI Encoder* is partially domain-dependent, validating (Left in Fig. 4) and training sets are from the same domain, but the testing set (Right in Fig. 4) is from a different unseen domain. As we can see, the disease-related features $v_{ma} + v_{mi}$ mainly reflect the disease-related information since they mainly reconstruct the lesion regions without mixing others. The disease-irrelevant $v_d$ features mainly learn features such as the contour of the breasts, pectoralis, and other irrelevant glands without lesion information. It is worth noting that the white dots on the image which are caused by machine shooting are learned by $v_d$ as visualization. This means that through the ability of domain generalization, our method can disentangle the irrelevant part successfully and prevent it from predicting the disease. Moreover, the macroscopic features $v_{ma}$ capture the macroscopic attributes of the lesions, *e.g.,* shape and density; while the microscopic features $v_{mi}$ learn properties like global context, texture, or other invisible features but related to disease classification. These results further indicate the effectiveness and interpretability of our DarMo.

## 5 CONCLUSION

We propose a novel **D**omain **A**gnostic **R**epresentation **Mo**del (DarMo) on domain generalization for medical diagnosis, in order to achieve robustness to multi centers. We evaluate our method on both public and in-house datasets. Potential results demonstrate the effectiveness of our DarMo, we will try to generalize this method to other medical imaging problems such as lung cancer, liver cancer, *etc*.

## 6 ACKNOWLEDGEMENT

This work was supported by MOST-2022ZD0114900, NSFC-62061136001, Hong Kong Research Grants Council through General Research Fund (Grant 17207722).

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

# A    FORMAL DESCRIPTION OF THEOREM 3.1

In this section, we present the formal version and the proof of theorem 3.1, which claims the *disentanglement* between disease-related features and center effects. In the following, we first introduce *model assumptions*, followed by *definition of disentanglement*; finally, we present the *formal version of theorem 3.1* and its proof.

***Model Assumptions and Notations.*** According to the causal graph in Fig. 2, the joint distribution over $(y, v_d, v_{mi}, v_{ma}, A, x)$ given each domain can be factorized as *conditional factors* Pearl (2009); Schölkopf et al. (2021):

$$p(y, v_d, v_{mi}, v_{ma}, A, x|d) = p(y)p(v_d|d)p(v_{mi}|y)p(v_{ma}|y)p(x|v_d, v_{mi}, v_{ma})p(A|v_{ma}).$$

In the following, we will introduce the assumption of each conditional factor. Specifically, for latent variables $v_d, v_{mi}, v_{ma}$, we assume that $v_{mi}|y$, $v_{ma}|y$ and $v_d|d$ belong to the following exponential families:

$$p(v_d|d) := p_{\mathbf{T}_d^v, \mathbf{\Gamma}_d^{v_d}}(v_d|d),\ p_{\mathbf{T}_{mi}^v, \mathbf{\Gamma}_y^{v_{mi}}}(v_{mi}|y),\ p_{\mathbf{T}^{v_{ma}}, \mathbf{\Gamma}_y^{v_{ma}}}(v_{ma}|y), \tag{3}$$

where

$$p_{\mathbf{T}^u, \mathbf{\Gamma}_o^u}(u|o) = \prod_{i=1}^{q_u} \exp\Big(\sum_{j=1}^{k_u} T_{i,j}^u(u_i)\Gamma_{o,i,j}^u + B_i(u_i) - C_{o,i}^u\Big), \tag{4}$$

for any $u \in \{v_{mi}, v_{ma}\}$ with $o = y$; and $u = v_d$ with $o = d$. The $\{T_{i,j}^u(u_i)\}$, $\{\Gamma_{o,i,j}^u\}$ denote the sufficient statistics and natural parameters, $\{B_i\}$ and $\{C_{o,i}^u\}$ denote the base measures and normalizing constants to ensure the integration of distribution equals to 1. Let $\mathbf{T}^u(u) := [\mathbf{T}_1^u(u_1), ..., \mathbf{T}_{q_u}^u(u_{q_u})] \in \mathbb{R}^{k_u \times q_u}$ $\big(\mathbf{T}_i^u(u_i) := [T_{i,1}^u(u_i), ..., T_{i,k_u}^u(u_i)], \forall i \in [q_u]\big)$, $\mathbf{\Gamma}_o^u := [\mathbf{\Gamma}_{o,1}^u, ..., \mathbf{\Gamma}_{o,q_u}^u] \in \mathbb{R}^{k_u \times q_u}$ $\big(\mathbf{\Gamma}_{o,i}^u := [\Gamma_{o,i,1}^u, ..., \Gamma_{o,i,k_u}^u], \forall i \in [q_u]\big)$. This assumption has been widely assumed in the literature of causal representation learning and causal learning Khemakhem et al. (2020); Sun et al. (2021).

For $x, A$, we assume the following *additive noise model* (ANM):

$$x = f_x(v_{mi}, v_{ma}, v_d) + \varepsilon_x, A = f_A(v_{ma}) + _A, \tag{5}$$

where $\varepsilon_x, \varepsilon_A$ denote the exogenous variables of $X$ and $A$, respectively.

***Definition of Disentanglement.*** With such model assumptions and formulations, we introduce our formal definition of *disentanglement*. First, we denote $\theta := \{\mathbf{T}^{v_{mi}}, \mathbf{T}^{v_d}, \mathbf{T}^{v_{ma}}, \mathbf{\Gamma}_y^{v_{mi}}, \mathbf{\Gamma}_y^{v_{ma}}, \mathbf{\Gamma}_d^{v_d}, f_x, f_A\}$ in the above models. We define the *disentanglement* as follows:

**Definition A.1** (Disentanglement of Latent Space). *We say that the $v_{mi}, v_{ma}, v_d$ are disentangled with each other under $\theta$, if for any $\tilde{\theta} := \{\tilde{\mathbf{T}}^{v_{mi}}, \tilde{\mathbf{T}}^{v_d}, \tilde{\mathbf{T}}^{v_{ma}}, \tilde{\mathbf{\Gamma}}_y^{v_{mi}}, \tilde{\mathbf{\Gamma}}_y^{v_{ma}}, \tilde{\mathbf{\Gamma}}_d^{v_d}, \tilde{f}_x, \tilde{f}_A\}$ that giving rise to the same observational distributions: $p_\theta(x, A, y|d) = p_{\tilde{\theta}}(x, A, y|d)$ for any $x, y, A$ and $d$, there exists invertible matrices $M_{v_{mi}}, M_{v_{ma}}, M_{v_d}$ and vectors $b_{v_d}, b_{v_{mi}}, b_{v_{ma}}$ such that:*

$$\tilde{\mathbf{T}}([\tilde{f}_x^{-1}]_{\mathcal{I}}(x)) = M_{v_{mi}}\mathbf{T}([f_x^{-1}]_{\mathcal{I}}(x)) + b_{v_{mi}}, \tag{6}$$

$$\tilde{\mathbf{T}}([\tilde{f}_x^{-1}]_{\mathcal{A}}(x)) = M_{v_{ma}}\mathbf{T}([f_x^{-1}]_{\mathcal{A}}(x)) + b_{v_{ma}}, \tag{7}$$

$$\tilde{\mathbf{T}}([\tilde{f}_x^{-1}]_{\mathcal{D}}(x)) = M_{v_d}\mathbf{T}([f_x^{-1}]_{\mathcal{D}}(x)) + b_{v_d}, \tag{8}$$

*where the $\mathcal{I}, \mathcal{A}, \mathcal{D}$ denote the space of the latent variables $v_{mi}, v_d, v_{ma}$. Correspondingly, for $f^{-1}(x)$ that transforms $x$ into the latent space $(\mathcal{I}, \mathcal{A}, \mathcal{D})$, $[f_x^{-1}]_{\mathcal{I}}(x), [f_x^{-1}]_{\mathcal{A}}(x)$ and $[f_x^{-1}]_{\mathcal{D}}(x)$ respectively denote the elements of $f^{-1}(x)$ in the space $\mathcal{I}, \mathcal{A}$ and $\mathcal{D}$.*

**Remark A.1.** *This definition is a variation of A-identifiability of Khemakhem et al. (2020) and the identifiability in Sun et al. (2021), which means the latent variables can be determined up to affine transformation with an invertible transformation matrix. Specifically, for any $x \leftarrow f_x(v_{mi}^*, v_{ma}^*, v_d^*)$, the $[f_x^{-1}]_{\mathcal{I}}(x), [f_x^{-1}]_{\mathcal{A}}(x)$ and $[f_x^{-1}]_{\mathcal{D}}(x)$ respectively return true latent variables $v_{mi}^*, v_{ma}^*, v_d^*$. Then if the model $\tilde{\theta}$ can perfectly fit the joint distribution over each domain, i.e., $p_\theta(x, A|d), [\tilde{f}_x^{-1}]_{\mathcal{I}}(x), [\tilde{f}_x^{-1}]_{\mathcal{A}}(x)$ and $[\tilde{f}_x^{-1}]_{\mathcal{D}}(x)$ can recover $v_{mi}^*, v_{ma}^*, v_d^*$ up to linear transformations with invertible matrices $M_{v_{mi}}, M_{v_{ma}}$ and $M_{v_d}$.*

***Formal Version of Theorem 3.1.*** We present the formal version of theorem 3.1 as follows:

**Theorem A.2** (Formal version of theorem 3.1). *Under the causal model in Fig. 2 with Eq. 3 and Eq. 5, for any $\theta$, we have that $v_{mi}, v_d, v_{ma}$ are disentangled, under following assumptions:*

1. *The characteristic functions of $\varepsilon_x$, $\varepsilon_A$ are almost everywhere nonzero.*

2. *$f_x, f_A$ are bijective functions;*

3. *The sufficient statistics are differentiable almost everywhere; besides, $\{T_{i,j}^u\}_{1 \leq j \leq k_u}$ are linearly independent in $\mathcal{I}$, $\mathcal{A}$ or $\mathcal{D}$ for each $i \in [q_u]$ for any $u = v_{mi}, v_{ma}, v_d$.*

4. *There exists $m$ different values of domain variable $d$, (i.e., $d_1, ..., d_m$) and $K$ different values of disease label $y$, (i.e., $y_1, ..., y_K$) such that $\left[[\boldsymbol{\Gamma}_{d_2}^{v_d} - \boldsymbol{\Gamma}_{d_1}^{v_d}]^\mathsf{T}, ..., [\boldsymbol{\Gamma}_{d_m}^{v_d} - \boldsymbol{\Gamma}_{d_1}^{v_d}]^\mathsf{T}\right]^\mathsf{T} \in \mathbb{R}^{m \times (q_{v_d} \times k_{v_d})}$ and $\left[[\boldsymbol{\Gamma}_{y_2}^{u=v_{mi},v_{ma}} - \boldsymbol{\Gamma}_{y_1}^{u=v_{mi},v_{ma}}]^\mathsf{T}, ..., [\boldsymbol{\Gamma}_{y_K}^{u=v_{mi},v_{ma}} - \boldsymbol{\Gamma}_{y_1}^{u=v_{mi},v_{ma}}]^\mathsf{T}\right]^\mathsf{T} \in \mathbb{R}^{m \times (q_u \times k_u)}$ have full column rank.*

**Remark A.2.** *These assumptions have been widely assumed in the literature of independent component analysis and representation learning Khemakhem et al. (2020); Sun et al. (2021); Li et al. (2021). Assumptions 1-3 is easy to satisfy. Specifically, the characteristic functions of $\varepsilon_x$ and $\varepsilon_A$ are almost everywhere non-zeros for most discrete (such as binomial, Poisson, geometric) continuous variables (such as Gaussian, student-t). For assumption 2, as it has been empirically verified Kramer (1991) that the extracted low dimensional embedding is able to recover the original image, it is natural for $f_x$ to be bijective. The bijectivity of $f_A$ is to ensure the disentanglement of $v_{ma}$ (up to affine transformation), as similarly adopted in Li et al. (2021).*

*For assumption 4, it is required that distribution across domains and disease status are diverse enough, which is easy to satisfy. Based on this assumption, $m$ (the number of domains) and $K$ (the number of disease statuses) are respectively required to be larger than the dimension of $v_d$, and the dimension of $(v_{mi}, v_{ma})$. This suggests that we should collect data from as many domains as possible, although empirically we find that three domains are enough to achieve disentanglement and generalization (as shown in Tab. 1 and Tab. 2 that out-of-domain performance is comparable to in-distribution performance). For disease label, we can access a more finer label, e.g. breast cancer stage D'Orsi et al. (2018), although empirically we find that binomial benign/malignancy label is able to disentangle disease-related features.*

*Proof.* For simplicity, we denote $\tilde{p}(u|o) := p_{\tilde{\mathbf{T}}^u, \tilde{\mathbf{\Gamma}}_o^u}(u|o)$. Since $p_\theta(x|d, y) = p_{\tilde{\theta}}(x|d, y)$, then we have

$$\int p_{f_x}(x|v_{mi}, v_{ma}, v_d) p(v_{mi}, v_{ma}|y) p(v_d|d) dv_{mi} dv_{ma} dv_d =$$

$$\int p_{\tilde{f}_x}(x|v_{mi}, v_{ma}, v_d) \tilde{p}(v_{mi}, v_{ma}|y) \tilde{p}(v_d|d) dv_{mi} dv_{ma} dv_d.$$

According to the chain rule of changing from $v_{mi}, v_{ma}, v_d$ to $\bar{x} := f_x(v_{mi}, v_{ma}, v_d)$, we have that $\int p_{\varepsilon_x}(x - \bar{x}) p(f_x^{-1}(\bar{x})|d, y) J_{f^{-1}}(\bar{x}) d\bar{x} = \int p_{\varepsilon_x}(x - \bar{x}) p(\tilde{f}_x^{-1}(\bar{x})|d, y) J_{\tilde{f}^{-1}}(\bar{x}) d\bar{x}$, where $J_f(x)$ denotes the Jacobian matrix of $f$ on $x$. Denote $p'(\bar{x}|d, y) := p(f_x^{-1}(\bar{x})|d, y) J_{f^{-1}}(\bar{x})$. Applying Fourier transformation to both sides, we have $F[p'](\omega) \varphi_{\varepsilon_x}(\omega) = F[\tilde{p}'](\omega) \varphi_{\varepsilon_x}(\omega)$, where $\varphi_{\varepsilon_x}$ denotes the characteristic function of $\varepsilon_x$. Since they are almost everywhere nonzero, we have that $F[p'](\omega) = F[\tilde{p}']$, which means that $p'(\bar{x}|d, y) = \tilde{p}'(\bar{x}|d, y)$. This is equivalent to the following:

$$\log \text{vol} J_{f_x}(x) + \sum_{u=v_{mi},v_{ma}} \sum_{i=1}^{q_u} (\log B_i([f_{x,i}^{-1}]_{\mathcal{U}}(x)) - \log C_i^u(y)$$

$$+ \sum_{j=1}^{k_u} T_{i,j}^u(f_{x,i}^{-1}(x)) \Gamma_{i,j}^u(x)) + \sum_{i=1}^{q_{v_d}} (\log B_i([f_{x,i}^{-1}]_{\mathcal{D}}(x))$$

$$- \log C_i^{v_d}(d) + \sum_{j=1}^{k_{v_d}} T_{i,j}^{v_d}(f_{x,i}^{-1}(x)) \Gamma_{i,j}^{v_d}(x)).$$

$$= \log \text{vol} J_{\tilde{f}_x}(x) + \sum_{u=v_{mi},v_{ma}} \sum_{i=1}^{q_u} (\log \tilde{B}_i([\tilde{f}_{x,i}^{-1}]_{\mathcal{U}}(x)) - \log \tilde{C}_i^u(y)$$

$$+ \sum_{j=1}^{k_u} \tilde{T}_{i,j}^u(\tilde{f}_{x,i}^{-1}(x))\tilde{\Gamma}_{i,j}^u(x)) + \sum_{i=1}^{q_{v_d}}(\log \tilde{B}_i([\tilde{f}_{x,i}^{-1}]_{\mathcal{D}}(x))$$

$$- \log \tilde{C}_i^{v_d}(d) + \sum_{j=1}^{k_{v_d}} \tilde{T}_{i,j}^{v_d}(\tilde{f}_{x,i}^{-1}(x))\tilde{\Gamma}_{i,j}^{v_d}(x)). \tag{9}$$

Subtract the Eq. 9 with $y = y_1$ from Eq. 9 with $y = y_k$ for $k \neq 1$, we have that

$$\sum_{u=v_{mi},v_{ma}}\left(\langle \mathbf{T}^u([f_x^{-1}]_{\mathcal{U}}(x)), \overline{\mathbf{\Gamma}}^u(y_k)\rangle + \sum_i \log \frac{C_i^u(y_1)}{C_i^u(y_k)}\right)$$

$$= \sum_{u=v_{mi},v_{ma}}\left(\langle \tilde{\mathbf{T}}^u([\tilde{f}_x^{-1}]_{\mathcal{U}}(x)), \overline{\tilde{\mathbf{\Gamma}}}^u(y_k)\rangle + \sum_i \log \frac{\tilde{C}_i^u(y_1)}{\tilde{C}_i^u(y_k)}\right), \tag{10}$$

for all $k \in [m]$, where $\overline{\mathbf{\Gamma}}(y) = \mathbf{\Gamma}(y) - \mathbf{\Gamma}(y_1)$. Denote $\tilde{b}_u(k) = \sum_{u=v_{mi},v_{ma}} \sum_i^{q_u} \frac{\tilde{C}_i^u(y_1)C_i^u(y_k)}{\tilde{C}_i^u(y_k)C_i^u(y_1)}$ for $k \neq 1$. Similarly, by subtracting the Eq. 9 with $d = d_1$ from Eq. 9 with $d = d_l$ for $l \neq 1$, we have

$$\langle \mathbf{T}^{v_d}([f_x^{-1}]_{\mathcal{D}}(x)), \overline{\mathbf{\Gamma}}^{v_d}(d_l)\rangle + \sum_i \log \frac{C_i^{v_d}(d_1)}{C_i^{v_d}(d_l)}$$

$$= \langle \tilde{\mathbf{T}}^{v_d}([\tilde{f}_x^{-1}]_{\mathcal{D}}(x)), \overline{\tilde{\mathbf{\Gamma}}}^{v_d}(d_l)\rangle + \sum_i \log \frac{\tilde{C}_i^{v_d}(d_1)}{\tilde{C}_i^{v_d}(d_l)}, \tag{11}$$

for all $k \in [m]$, where $\overline{\mathbf{\Gamma}}(d) = \mathbf{\Gamma}(d) - \mathbf{\Gamma}(d_1)$. Denote $\tilde{b}_{v_d}(l) = \sum_i \frac{\tilde{C}_i^{v_d}(d_1)C_i^{v_d}(d_l)}{\tilde{C}_i^{v_d}(d_l)C_i^{v_d}(d_1)}$ for $l \neq 1$, we have that:

$$\overline{\mathbf{\Gamma}}^{v_d,\top}\mathbf{T}^{v_d}([f_x^{-1}]_{\mathcal{D}}(x)) = \overline{\tilde{\mathbf{\Gamma}}}^{v_d,\top}\tilde{\mathbf{T}}^{v_d}([\tilde{f}_x^{-1}]_{\mathcal{D}}(x)) + \tilde{b}_{v_d}, \tag{12}$$

$$\overline{\mathbf{\Gamma}}^{v_{mi},\top}\mathbf{T}^{v_{mi}}([f_x]_{\mathcal{I}}^{-1}(x)) + \overline{\mathbf{\Gamma}}^{v_{ma},\top}\mathbf{T}^{v_{ma}}([f_x]_{\mathcal{A}}^{-1}(x))$$

$$= \overline{\tilde{\mathbf{\Gamma}}}^{v_{mi},\top}\tilde{\mathbf{T}}^{v_{mi}}([\tilde{f}_x]_{\mathcal{I}}^{-1}(x)) + \overline{\tilde{\mathbf{\Gamma}}}^{v_{ma},\top}\tilde{\mathbf{T}}^{v_{ma}}([\tilde{f}_x]_{\mathcal{A}}^{-1}(x)) + \tilde{b}_{v_{ma}} + \tilde{b}_{v_{mi}}. \tag{13}$$

Similarly, we also have $p'(\bar{A}|y) = \tilde{p}'(\bar{A}|y)$, which means that

$$\log \mathrm{vol} J_{f_A}(A) + \sum_{i=1}^{q_{v_{ma}}}(\log B_i([f_A^{-1}]_{\mathcal{A},i}(A)) - \log C_i^{v_{ma}}(d)$$

$$+ \sum_{j=1}^{k_{v_{ma}}} T_{i,j}^{v_{ma}}([f_A^{-1}]_{\mathcal{A},i}(A))\Gamma_{i,j}^{v_{ma}}(x))$$

$$= \log \mathrm{vol} J_{\tilde{f}_A}(A) + \sum_{i=1}^{q_{v_{ma}}}(\log B_i([\tilde{f}_A^{-1}]_{\mathcal{A},i}(A)) - \log C_i^{v_{ma}}(d)$$

$$+ \sum_{j=1}^{k_{v_{ma}}} \tilde{T}_{i,j}^{v_{ma}}([\tilde{f}_A^{-1}]_{\mathcal{A},i}(A))\tilde{\Gamma}_{i,j}^{v_{ma}}(A)), \tag{14}$$

which has that

$$\overline{\mathbf{\Gamma}}^{v_{ma},\top}\mathbf{T}^{v_{ma}}([f_A^{-1}]_{\mathcal{A}}(A)) = \overline{\tilde{\mathbf{\Gamma}}}^{v_{ma},\top}\tilde{\mathbf{T}}^{v_{ma}}([\tilde{f}_A^{-1}]_{\mathcal{A}}(A)) + \tilde{b}_{v_{ma}}. \tag{15}$$

Denote $v := [x^\top, A^\top]^\top$, $\varepsilon := [\varepsilon_x^\top, {}_A^\top]^\top$, $h(v) = [[f_x]_{\mathcal{I}}^{-1}(x)^\top, [f_A^{-1}]_{\mathcal{A}}(A)^\top]^\top$. Applying the same trick above, we have that

$$\overline{\mathbf{\Gamma}}^{v_{ma},\top}\mathbf{T}^{v_{ma}}([f_x]_{\mathcal{A}}^{-1}(x)) = \overline{\tilde{\mathbf{\Gamma}}}^{v_{ma},\top}\tilde{\mathbf{T}}^{v_{ma}}([\tilde{f}_x]_{\mathcal{A}}^{-1}(x)) + \tilde{b}_{v_{ma}}. \tag{16}$$

Combining Eq. 12, 13, 16, we have that

$$\overline{\mathbf{\Gamma}}^{v_d,\top}\mathbf{T}^{v_d}([f_x^{-1}]_{\mathcal{D}}(x)) = \overline{\tilde{\mathbf{\Gamma}}}^{v_d,\top}\tilde{\mathbf{T}}^{v_d}([\tilde{f}_x^{-1}]_{\mathcal{D}}(x)) + \tilde{b}_{v_d}, \tag{17}$$

$$\overline{\mathbf{\Gamma}}^{v_{ma},\top}\mathbf{T}^{v_{ma}}([f_x]_{\mathcal{A}}^{-1}(x)) = \overline{\tilde{\mathbf{\Gamma}}}^{v_{ma},\top}\tilde{\mathbf{T}}^{v_{ma}}([\tilde{f}_x]_{\mathcal{A}}^{-1}(x)) + \tilde{b}_{v_{ma}}. \tag{18}$$

$$\overline{\mathbf{\Gamma}}^{v_{mi},\top}\mathbf{T}^{v_{mi}}([f_x]_{\mathcal{I}}^{-1}(x)) = \overline{\tilde{\mathbf{\Gamma}}}^{v_{mi},\top}\tilde{\mathbf{T}}^{v_{mi}}([\tilde{f}_x]_{\mathcal{I}}^{-1}(x)) + \tilde{b}_{v_{mi}}. \tag{19}$$

Applying the same trick in (Sun et al., 2021, Theorem 7.9) due to assumption *3, 4*, we have that $(\overline{\mathbf{\Gamma}}^{u,\top})^{-1}\overline{\tilde{\mathbf{\Gamma}}}^{u,\top}$ are invertible for $u = v_{mi}, v_{ma}, v_d$. The proof is completed by setting $M_u, b_u$ in Def. A.1 as $(\overline{\mathbf{\Gamma}}^{u,\top})^{-1}\overline{\tilde{\mathbf{\Gamma}}}^{u,\top}$ and $\tilde{b}_u$ for $u = v_{mi}, v_{ma}, v_d$. $\qquad\square$

# B  OBJECTIVE FUNCTION

## B.1  FINAL LOSS

Our final loss function is the summation of the loss in Eq. 1, *i.e.*, $\sum_d \ell^d \sum_d \ell^d(q^d, p_\theta^d)$, where each $\ell^d(q^d, p_\theta^d)$ is:

$$\ell^d(q^d, p_\theta^d) = -\frac{1}{n_d}\sum_{i=1}^{n_d}\left(\log p_\theta^d(y_i, A_i|x_i) + \mathbb{E}_{q_{\psi^d}(v|x_i)}\left(\log\frac{p_\theta^d(x_i, v)}{q_{\psi^d}(v|x_i)}\right)\right),$$

$$= \underbrace{-\frac{1}{n_d}\sum_{i=1}^{n_d}\left(\log\left(\mathbb{E}_{q_{\psi^d}(v|x_i)}\left(p_\theta(A_i|v_{ma})p_\theta(y_i|v_{ma}, v_{mi})\right)\right)\right)}_{\text{prediction of } A, y}$$

$$\underbrace{-\frac{1}{n_d}\sum_{i=1}^{n_d}\mathbb{E}_{q_{\psi^d}(v|x_i)}\left(\log p_\theta(x_i|v)\right)}_{\text{reconstruction loss}} \underbrace{-\frac{1}{n_d}\sum_{i=1}^{n_d}\text{KL}(q_{\psi^d}(v|x_i), p_\theta^d(v))}_{\text{KL divergence}}.$$

The first term is to the cross entropy loss for $A$ and $y$; for each sample $x_i$, we first generate $v_{mi}, v_{ma}$ from $x_i$ via $q_{\psi^d}(v|x_i)$, then feed $v_{mi}, v_{ma}$ into $p_\theta(y_i|v_{ma}, v_{mi})$ and $v_{ma}$ into $p_\theta(A_i|v_{ma})$ to predict $y$ and $A$, respectively. The second and third terms are respectively reconstruction loss and KL divergence loss in VAE.

## B.2  DERIVATION OF OUR OBJECTIVE

The log-likelihood over the observations $(x, y, A)$ in the Bayesian network is given by:

$$\log p(x, y, A; \theta) = \log p(x; \theta) + \log p(A|x; \theta) \\ + \log p(y|x, A; \theta) \tag{20}$$

which forms the learning objective of our problem. Next, we give the details about how the loss functions for optimization are derived from the likelihood.

For the log-likelihood $\log p(x; \theta)$ of each domain, we have the ELBO as a lower bound on the log-likelihood:

$$\log p(x; \theta) = KL(q(v_d, v_{mi}, v_{ma}|x)||p(v_d, v_{mi}, v_{ma}|x)) + \\ \mathbb{E}_{q(v_d, v_{mi}, v_{ma}|x)}\log p(v_d, v_{mi}, v_{ma}, x) - \mathbb{E}_{q(v_d, v_{mi}, v_{ma}|x)}\log q(v_d, v_{mi}, v_{ma}|x) \\ \geq \mathbb{E}_{q(v_d, v_{mi}, v_{ma}|x)}\log\left(\frac{p(v_d, v_{mi}, v_{ma}, x)}{q(v_d, v_{mi}, v_{ma}|x)}\right) \\ = -KL(q(v_d, v_{mi}, v_{ma}|x)||p(v_d, v_{mi}, v_{ma})) \\ + \mathbb{E}_{q(v_d, v_{mi}, v_{ma}|x)}\log\left(p(x|v_d, v_{mi}, v_{ma})\right) \tag{21}$$

where $q(\cdot|x)$ denotes $q(v_d, v_{mi}, v_{ma}|x)$ for simplicity. Specifically, we use $\theta$ to parameterize $p(v_d, v_{mi}, v_{ma}, x)$ and $\phi$ to parameterize $q(v_d, v_{mi}, v_{ma}|x)$. The prior joint distribution

$p_\theta(v_d, v_{mi}, v_{ma}, x)$ can be factorized into $p_\theta^d(z)p_\theta(v_{mi}, v_{ma})p_\theta^d(x|v_d, v_{mi}, v_{ma})$. Under mean-field approximation, the posterior $q_\phi(v_d, v_{mi}, v_{ma}|x)$ can be factorized into $q_\phi^d(v_d|x)q_\phi(v_{mi}, v_{ma}|x)$. Note that the index $d$ is added since $v_d$ and $x$ are domain-variant and $v_{mi}, v_{ma}$ are domain-invariant. The final two terms in Eq. 21 are the KL loss and reconstruction loss in the loss functions.

For the conditional log-likelihood $\log p(A|x; \theta)$, we have:

$$\log p(A|x; \theta) = \log \int p_\theta(A|v_{ma})p_\theta(v_{ma}|x)dv_{ma} \tag{22}$$

where $p_\theta(v_{ma}|x)$ is re-parameterized by the posterior model $q_\phi(v_{ma}|x)$ in the variational framework above. Under one-time sampling, we have $\log p(A|x; \theta) = \log p_\theta(A|v_{ma})p_\theta(v_{ma}|x)$. Since the different attributes are independent in $A$, and each attribute $\{g_i\}_{i\in[C]} \in A$ ($[C] := \{1, ..., C\}$) obeys binomial distribution, we can rewrite the log-likelihood as:

$$\begin{aligned}
\log p(A|x; \theta) &= \log p(g_1, \cdots, g_C|x; \theta) \\
&= \log \prod_{i=1}^{C} p(g_i|x; \theta) \\
&= \log \prod_{i=1}^{C} \hat{g}_i^{g_i}(1 - \hat{g}_i)^{1-g_i} \\
&= \sum_{i=1}^{C} g_i \log \hat{g}_i + (1 - g_i) \log(1 - \hat{g}_i)
\end{aligned} \tag{23}$$

where $\hat{g}_i$ denotes the probability of that the sample $x$ contains attribute $i$ (*i.e.*, $g_i$ being 1 under the prediction of $p_\theta(A|v_{ma})q_\theta(v_{ma}|x)$). Thus we derive the multi-label loss for the Graph Convolutional Network.

For the conditional log-likelihood $\log p(y|x, A; \theta)$, we have:

$$\log p(y|x, A; \theta) = $$
$$\log \int p_\theta(y|v_{mi}, v_{ma})p_\theta(v_{mi}, v_{ma}|x, A)dsda \tag{24}$$

where $p_\theta(v_{mi}, v_{ma}|x, A)$ is re-parameterized by the posterior model $q_\phi(v_{mi}, v_{ma}|x)$ in the variational framework above. Under one-time sampling, we have $\log p(y|x, A; \theta) = \log p_\theta(y|v_{mi}, v_{ma})q_\theta(v_{mi}, v_{ma}|x)$. Since $y$ obeys binomial distribution, we can rewrite the log-likelihood as:

$$\begin{aligned}
\log p(y|x, A; \theta) &= \log \hat{y}^y(1 - \hat{y})^{(1-y)} \\
&= y \log \hat{y} + (1 - y) \log(1 - \hat{y})
\end{aligned} \tag{25}$$

where $\hat{y}$ denotes the probability of y being 1 under the prediction of $p_\theta(y|v_{mi}, v_{ma})q_\theta(v_{mi}, v_{ma}|x)$. Thus we derive the loss function for the binary classification of benign/malignant.

## C  ABLATION STUDY ON EACH COMPOMENT

Here are some interpretations for the variants: **a) *DADI*** denotes whether using *DADI encoder* during the reconstructing phase, while *DAL* denotes using domain-aware layers for distinguishing multiple domains in *DADI encoder*; **b) *Attribute Learning*** denotes the way to predict attributes: $\times$ means no predictions of attributes, *multi-task* means using a fully connected layer to predict the multiple attributes, and $\mathcal{L}_{gcn}$ means using our Disease-Attribute Generative Model to predict attributes; **c)** $v_{mi}$ denotes whether split the latent factor $v_{mi}$ out for disentanglement in training; **d) *Medical Image Decoder*** denotes whether use the reconstruction loss in training.

As shown in Tab. 4, every component is effective. It is worth noting that using naive GCN also leads to a boosting of around 5% in average. Such a result can demonstrate that the attributes can guide the

Table 4: Ablation Study: AUC evaluation of public/in-house datasets on **external cohorts** (unseen domains, *i.e.*, out-of-distribution: training and testing data are from different domains, testing on InH1/InH2/InH3/DDSM while training on the other three). Testing names are noted in the table.

| DADI | Attribute Learning | $v_{mi}$ | Medical Image Decoder | **InH1** | **InH2** | **InH3** | **DDSM** |
|---|---|---|---|---|---|---|---|
| × | × | × | × | 0.822 | 0.758 | 0.735 | 0.779 |
| × | multi-task | × | × | 0.851 | 0.793 | 0.775 | 0.801 |
| × | $\mathcal{L}_{gcn}$ | × | × | 0.877 | 0.827 | 0.804 | 0.830 |
| × | $\mathcal{L}_{gcn}$ | × | ✓ | 0.911 | 0.846 | 0.816 | 0.844 |
| DAL | $\mathcal{L}_{gcn}$ | × | ✓ | 0.931 | 0.862 | 0.841 | 0.878 |
| DAL | × | × | ✓ | 0.913 | 0.840 | 0.823 | 0.852 |
| × | $\mathcal{L}_{gcn}$ | ✓ | ✓ | 0.916 | 0.851 | 0.821 | 0.859 |
| DAL | $\mathcal{L}_{gcn}$ | ✓ | ✓ | **0.948** | **0.874** | **0.858** | **0.892** |

Table 5: Ablation study on the combination of training data sets. Take testing on the public dataset DDSM as an example. (OOD settings)

| train on InH(1,2) | train on InH(1,3) | train on InH(3,2) | train on InH(1,2,3) |
|---|---|---|---|
| 0.885 | 0.881 | 0.887 | 0.892 |

learning of disease-related features. Meanwhile, disentanglement learning also causes a noticeable promotion, which may be due to that the disease-related features can be easier identified through disentanglement learning without mixing information with others. Moreover, Lines 5-6 validate that disease-related features can be disentangled better with the guidance of exploring attributes. Lines 7-8 validate that distinguishing multiple domains improves the generalization performance.

# D  MORE ABLATION STUDY

We also explore the impact of the combination of training domains and try different training combinations for unseen test domains. Take testing on DDSM (Bowyer et al., 1996) as an example. As shown in Tab. 5, the more types of domains the better effect of our model. Due to the different correlations between different domains, the effect will be different under different combinations. But based on the inter mechanism of our model, influences between different domains are not obvious and three domains are sufficient to achieve comparable results.

Under the setting: testing on DDSM (Bowyer et al., 1996) (OOD) while training on InH1+InH2+InH3, we also list the results of our invariant model DarMo (OOD model) under testing on the testing set of InH1/InH2/InH3 (in-distribution) as shown in Tab. 6. We also do experiments under the setting: training on InH1+InH2+InH3+DDSM and testing on InH1/InH2/InH3/DDSM. In addition, under the same setting, we also test our variant model **Ours-single** (in-distribution model). The results of testing on unseen DDSM (Bowyer et al., 1996) (OOD) is 0.861, testing on InH1/InH2/InH3 (in-distribution) which are from the same training sets (InH1+InH2+InH3) are 0.944, 0.880, and 0.853 respectively. The variant model testing on InH1/InH2/InH3 (the same domain as the training set) behaves comparably with ours in Tab. 6 and is slightly better since our DarMo split some inter-domain correlation which can decent performance under domain generalization. Thus, the variant model faces a larger drop over our invariant model DarMo when generalizing to the unseen DDSM dataset (0.892 vs 0.861).

# E  MORE DETAILS OF DISEASE ATTRIBUTE GENERATIVE MODELS.

To capture macroscopic patterns in $v_{ma}$, we constrain it to estimate the clinical attributes $A$ that include macroscopic information such as shape, margins, lobulation, *etc.* Besides, we constrain it and $v_{mi}$ to predict the disease label $y$, with $v_{mi}$ accounting for additional microscopic information of lesions. We note that such constraints align with the causal graph in Fig. 2, as only $v_{ma} \rightarrow A$ and $y \perp v_c | v_{ma}, v_{mi}$. Finally, we constrain all factors to reconstruct the input $x$, with $v_d$ responsible for the domain-aware effects in $x$ (Medical Image Decoder ). Indeed, such asymmetric roles of $v_{ma}, v_{mi}, v_d$ in terms of relations with $y, A, x$ can additionally help to disentangle them from

Table 6: AUC of testing on data set InH1/InH2/InH3/DDSM while training on InH1+InH2+InH3.

| test on InH1 | test on InH2 | test on InH3 | test on DDSM |
|---|---|---|---|
| 0.939 | 0.874 | 0.852 | 0.892 |

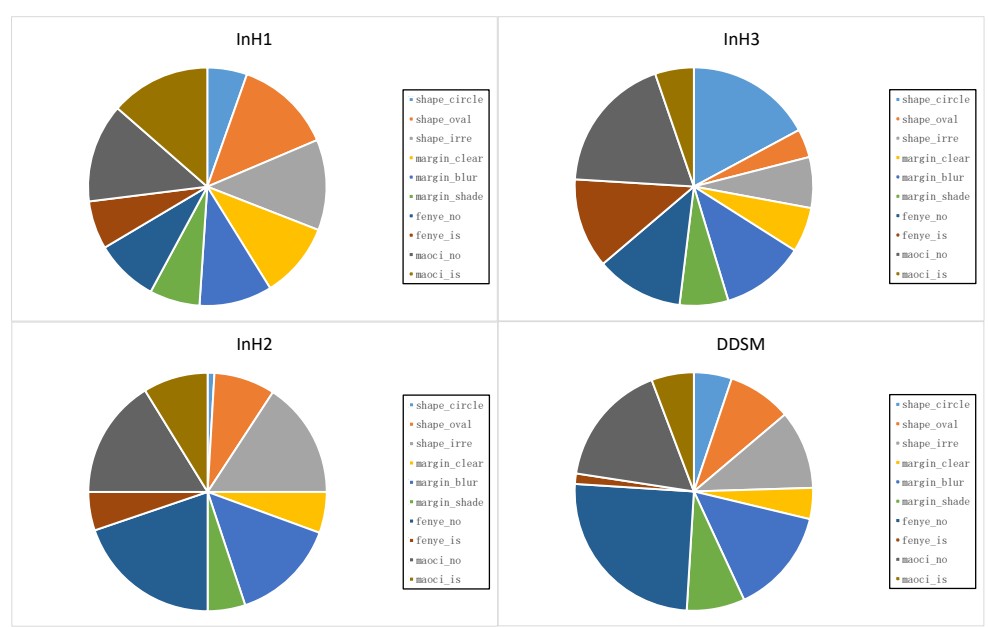

Figure 5: Distribution of lesions' characteristics in each center (dataset).

each other, on the basis of the two-branch encoder. We parameterize $p_\theta(y, A, x|v)$ as $p_{\theta_x}(x|v)$, $p_{\theta_y}(y|v_{ma}, v_{mi})$ and $p_{\theta_A}(A|v_{ma})$.

To utilize these relations, we parameterize the $p_\theta(A|v_{ma})$ by a Graph Convolutional Network (GCN) which is a flexible way to capture the topological structure in the label space. Along with (Chen et al., 2019), we build a graph $G = (V, E)$ with twelve nodes and consider each attribute as a node, *e.g.,* Shape-circle, Margin-clear. Each node $V_i \in V$ represents the word embedding of the attributes. Each edge $e \in E$ represents the inter-relevance between attributes. The inputs of the graph are feature representations $H^l$ and corresponding correlation matrix $B$ which is calculated in the same way Wang et al. (2021b). For the first layer, $H^0 \in \mathbb{R}^{c \times c'}$ denotes the one-hot embedding matrix of each attribute node where $c$ is the number of attributes, $c'$ is the length of embeddings. Then, the feature representation of the graph at every layer (Kipf & Welling, 2016) can be calculated via $H^{l+1} = \delta(BH^lW^l)$, where $\delta(\cdot)$ is LeakyRelu (Maas et al., 2013), $W^l$ is the transformation matrix which is the parameter to be learned in the $l$th layer. The output $\{\hat{A}^k\}_k = \text{GCN}([\text{Causal-Encoder}(x)]_A))$ is learned to approximate attributes $\{A^k\}_k$.

## F    MORE DETAILS OF IMPLEMENTATION AND DATASETS.

External cohorts are unseen before testing, *i.e.*, have not been used in the training phase. For each dataset, the region of interest (ROIs) (malignant/benign masses) are cropped based on the annotations of radiologists the same as Kim et al. (2018). The training/valid/testing samples we use contain 1165 ROIs from 571 patients/143 ROIs from 68 patients/147 ROIs from 75 patients in DDSM (Bowyer et al., 1996), 684 ROIs from 292 patients/87 ROIs from 38 patients/83 ROIs from 33 patients in InH1, 840 ROIs from 410 patients/104 ROIs from 50 patients/105 ROIs from 52 patients in InH2, and 565 ROIs from 271 patients/70 ROIs from 33 patients/70 ROIs from 34 patients in InH3. The distribution of lesions' characteristics in each center (dataset) we use is shown in Fig. 5. And the distribution of ages in each center we use is shown in Fig. 6.

For a fair comparison, all methods are conducted under the same setting and share the same encoder backbone, *i.e.*, ResNet34 (He et al., 2016). Meanwhile, the decoder is the deconvolution network of

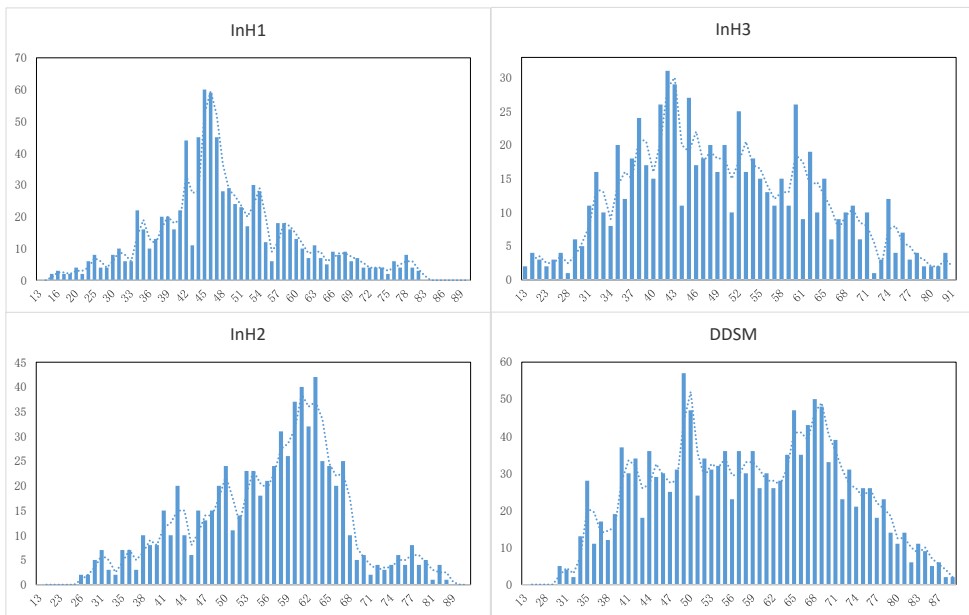

Figure 6: Distributions of ages in each center (dataset).

the encoder. For attribute annotations, in DDSM (Bowyer et al., 1996) annotations can be parsed from the ".OVERLAY" file. The third line in the ".OVERLAY" file has annotations for types, shapes, and margins of masses. And in our in-house datasets, we obtain attribute annotations from the verification of one director doctor based on the annotations of three senior doctors. We implement all models with PyTorch. We implement Adam for optimization. The weight hyperparameter in *variance regularizer* $\beta$ is 1 in our experiments.

The clinical attributes contain *circle, oval, irregular, circumscribed, obscured, ill-defined, is-lobulated, not-lobulated, is-spiculated, not-spiculated*. We add additional benign and malignant nodes to learn the correlation between the combination of attributes and benign/malignant. For the implementation of compared baselines, we directly load the published codes of ERM (He et al., 2016), Chen *et al.* (Chen et al., 2019), DANN (Ganin et al., 2016), MMD-AAE (Li et al., 2018), DIVA (Ilse et al., 2020), IRM (Arjovsky et al., 2019) and Prithvijit *et al.*(Chattopadhyay et al., 2020) during the test stage; while we re-implement methods of Guided-VAE (Ding et al., 2020b), ICADx (Kim et al., 2018) and Li *et al.* (Li et al., 2019) for lacking published source codes.

## G    TEST SET OF DDSM

We use the same To provide convenience for the latter works, we publish the list of our test division on the public dataset DDSM (Bowyer et al., 1996).

```
benign_12_case1889   benign_04_case3357   cancer_01_case3084
benign_04_case0304   benign_09_case4060   benign_05_case1491
cancer_08_case1464   cancer_09_case0049   cancer_11_case1678
cancer_04_case1090   cancer_05_case0157   benign_06_case0366
benign_04_case0270   benign_02_case1321   cancer_05_case0142
cancer_05_case0127   benign_04_case3103   cancer_07_case1143
cancer_08_case1128   benign_11_case1792   benign_06_case0396
cancer_15_case3371   benign_07_case1686   benign_13_case0485
benign_09_case4085   cancer_02_case0112   cancer_15_case3398
benign_03_case1435   cancer_01_case3027   cancer_07_case1114
cancer_03_case1070   benign_03_case1432   cancer_06_case1182
cancer_05_case0140   benign_12_case1947   benign_12_case1922
cancer_05_case0210   cancer_08_case1403   cancer_05_case0173
benign_01_case0235   benign_02_case1317   benign_11_case1836
cancer_05_case0222   cancer_08_case1532   benign_06_case0372
```

```
cancer_02_case0077  benign_11_case1855  cancer_05_case0139
benign_08_case1786  cancer_07_case1159  cancer_10_case1573
cancer_05_case0181  benign_09_case4038  cancer_05_case0192
benign_06_case0363  cancer_06_case1122  benign_01_case3113
benign_09_case4003  benign_06_case0367  cancer_12_case4139
cancer_14_case1985  cancer_05_case0183  cancer_10_case1642
cancer_05_case0206  cancer_03_case1007  cancer_12_case4108
cancer_09_case0340  benign_07_case1412  cancer_05_case0085
benign_09_case4065  benign_03_case1363  benign_09_case4027
benign_10_case4016  benign_13_case3433  benign_09_case4090
```

