# OpenReview forum: "Learning Domain-Agnostic Representation for Disease Diagnosis"
_ICLR.cc/2023/Conference — ICLR 2023 poster_

### Official Review · Reviewer_kDRU · 2022-10-17

**Confidence:** 3
**Correctness:** 3
**Technical Novelty And Significance:** 3
**Empirical Novelty And Significance:** 4
**Recommendation:** 8

**Clarity, Quality, Novelty And Reproducibility:**

Clear introduction of the model with a well-motivated problem. While the model is complex, the authors did a good job of describing it in detail. Only the maths could be clarified.

The paper tackles the important problem of robustness when models are transferred between centres. The approach is an elegant solution to disentangle the centre-specific characteristics from the features predictive of the outcome of interest.

Note on reproducibility: it is not mentioned if the data and code will be made publicly available, which may endanger the work reproducibility. As the trained model and data may disclose patients' data, I think only the model's code could and should be released due to the high model's complexity.

**Strength And Weaknesses:**

The paper clearly presents the motivation and the studied problem, which is highly relevant to the medical community. The paper is well structured and written which makes the proposed model clear even if complex. Moreover, the authors propose relevant ablation studies and thorough evaluation against state-of-the-art methodologies.

The main weakness of this work is the mathematical proofs, particularly the one in Appendix. Specifically, the theorem in the main text is not quite precise, nor particularly intuitive. It is also not easy to connect with the provided proof (the comment following 3.1 is much more valuable). The notations (such as log vol) should also be introduced and simplified as much as possible. References to equations should be numbered, instead of ‘the one with y_k’, ‘the one with d_t’ which makes it difficult to follow.

Minor updates that could improve the paper:
- Displaying confidence in the tables (it is currently not clear if the results are significantly better)
- Adding in Appendix a robustness metric, i.e. the difference in performance when trained in the same set and transferred. The performances should be computed on exactly the same test set.
- Adding reference to graph convolutional network (also it is referenced once as 'graph convolution network'),



**Summary Of The Paper:**

This paper proposes a variational autoencoder strategy to offer robustness to hospital-specific X-ray characteristics. The methodology aims to disentangle these centre-specific measurement characteristics from the features predictive of the condition. The paper presents theoretical intuition and evidence of the stronger robustness of malignancy classification on 4 mammogram datasets by demonstrating how their model trained on a subset of these datasets generalizes better to a dataset external to the training set.

**Summary Of The Review:**

Good paper, clear with great experimental and technical work. Would benefit from clearer proofs of the main theoretical result.

---

### Official Review · Reviewer_RBtC · 2022-10-24

**Confidence:** 3
**Correctness:** 3
**Technical Novelty And Significance:** 2
**Empirical Novelty And Significance:** 3
**Recommendation:** 6

**Clarity, Quality, Novelty And Reproducibility:**

Their idea of learning disentangled features in terms of domain effects is interesting enough for the potential reader. Also, they showed results from the prior works by training and testing on their datasets. But, with more experiments suggested above, their claim can be more strengthened.

**Strength And Weaknesses:**

Strength
•The authors implemented lots of prior works as baselines and evaluated their performances using the public and three in-house datasets to compare with their proposed model.
•The main problem they are interested in, causal relationships between domain, macroscopic, and microscopic dependent features, and their proposed model based on VAE are well described with appropriate example images, sufficient explanations, informative diagrams, and equations.
•They performed ablation studies to verify the effectiveness of each component in their DarMo, which helped the reviewer understand why each element is needed.
•They shared visualizations of reconstructions of latent variables to show that their proposed model learned disentangled features.
Weakness
•Can you elaborate how the domain-aware BN layer is implemented? If there are 4 domains (centers), each center image is fed into each layer of DADI encoder (that means 4 channels)? Is the BN layer just the regular BN layer without modification?
•In Figure 3, the latent variables extracted from DADI encoder are just combined with the features extracted from DADR encoder to generate input images using “medical image decoder”. The reviewer is not sure how the domain-aware disease-irrelevant features affect the classification of benign and malignant masses which is the main output of this model.
•Can the authors provide more details on how the four datasets are different in terms of domain effects including imaging acquisition, manufacturer, patient population, etc. It would be more interesting to see visualizations for each domain effect.
•Is the input data to the model a mammogram image or a patch of lesion region? Are the evaluation based on study level or image level? If all the results reported in this study are image-level, the reviewer would suggest that study-level results are better to be used for comparisons as study-level interpretation are usually needed in a clinical setting.
•In addition to the empirical results, it would be good to show an analysis on disentanglement using simplified datasets like different manufacturers including Hologic and GE machines. With DarMo, the domain-specific features (of each manufacturer) can be visualized and more informative to the potential reader. Visualization results shown in Figure 4 do not give the reader what domain-aware features are learned.
•The reviewer might miss it, but did the authors evaluate performance of conventional CNNs (like ResNet50) as results shown in Table 1? Also, it would be good to show performance of model trained on all datasets (InH1+InH2+InH3+DDSM) and evaluated on each dataset. If their proposed model performance is comparable or superior to it, it would be nice to demonstrate the robustness of the proposed model to distributional shift.


**Summary Of The Paper:**

In this study, the authors proposed Domain Agnostic Representation Model (DarMo) based on variational auto encoder, a learning framework to disentangle disease-related features from domain-related (center-related) effects to achieve robust prediction. They claimed that their work is the first to prove the disentanglement is feasible and their model can achieve state-of-the-art performance in terms of robustness to distributional shift for breast cancer detection using a public (DDSM) and three in-house datasets.

**Summary Of The Review:**

In general, their work and claim are well explained and verified with experimental results, ablation studies, and visualizations. However, some additional experiments might be needed to improve the quality of their study.

---

### Official Review · Reviewer_eNnv · 2022-10-26

**Confidence:** 3
**Correctness:** 3
**Technical Novelty And Significance:** 3
**Empirical Novelty And Significance:** 3
**Recommendation:** 6

**Clarity, Quality, Novelty And Reproducibility:**

Clarity: The paper is well organized. Most parts of the paper are written in a clear way.

Quality: The quality of the paper can be further improved.

Novelty: The novelty is fair.

Originality: Good.

Reproducibility: There is not source codes provided.


**Strength And Weaknesses:**

S1: The methodology part is written in a clear way and well addressed the main claims in the introduction part.

S2: It is an interesting idea to apply structural causal modeling to encode medical
prior knowledge and to disentangle disease-related features from the domain effect.


W1: It is better to provide complexity analysis. On top of accuracy and AUC, time efficiency should be considered in practical application.

W2: It is better to involve more components, e.g., loss functions, in Section 3.2 to make it clearer.

W3: Sun et al., 2021a is duplicated.

W4: The source codes are not available to validate the experimental results.


**Summary Of The Paper:**

This paper proposes a model DarMo to disentangle disease-related features from domain-related effects to achieve robust prediction in the healthcare domain.
The proposed model is guided by structural causal modeling to explicitly model disease-related and center effects.
Experiments on four real-world datasets demonstrate the effectiveness of DarMo.


**Summary Of The Review:**

Above the acceptance threshold.
Please see Strength And Weaknesses for detail.

---

### Decision · Program_Chairs · 2023-01-20

**Decision:**

Accept: poster

**Justification For Why Not Higher Score:**

The basic method of the paper is based on an existing VAE method for ICA. The math in the appendix could be more precise. The assumption that the domain will not affect the pixels of the disease seems a little unrealistic, since different hospitals may have different equipment.

**Justification For Why Not Lower Score:**

Domain-robustness is an important topic in medical imaging diagnostics. The paper is does a good job w.r.t. theory and experimental results (needs better formalism though). Reviewers were all positive. Rebuttal helped improve the paper.

**Metareview: Summary, Strengths And Weaknesses:**

The work proposes a method for creating domain-robust representations for medical images using a factor causal DAG. The method tackles the important problem of robustness when transferring models between hospitals or domains. The approach uses VAE methods for ICA decomposition to find a disentangled subspace. The reviewers agree that the proposed approach is reasonable, but note that the proofs and discussion in the paper are informal and could be improved. Some equation references do not point to the correct equations.

- The disentangled representation method is based on a factor causal DAG. The causal DAG seems reasonable (domains may change pixels related to disease, but the paper's assumption is OK).
- The reviewers agree the work tackles the important problem of robustness when models are transferred between hospitals (domains).
- The task is mostly reduced to finding a subspace that is disentangled (provided enough domains and classes w.r.t. the dimension of the input image).
- The proposed approach uses VAE methods for ICA decomposition. The proofs improved in the rebuttal but a lot of the discussion is still a little too informal (some equation references point to equations that we will only see in the future).

**Note From Pc:**

if the above contains the word "oral" or "spotlight" please see: "oral" presentation means -> notable-top-5% and "spotlight" means -> notable-top-25%. As stated in our emails, we are disassociating presentation type from AC recommendations